# Van der Waals Equation for the Description of Monolayer Formation on Arbitrary Surfaces

**Volodymyr Kutarov [1] and Eva Schieferstein [2],***

[1]  Research Institute of Physics, I.I. Mechnikov National University, 27 Paster Str., 65082 Odessa, Ukraine; v.kutarov@onu.edu.ua
[2]  Fraunhofer UMSICHT, Osterfelder Str. 3, D-46047 Oberhausen, Germany
*   Correspondence: eva.schieferstein@umsicht.fraunhofer.de

**Abstract:** The van der Waals equation is well known for the description of two-dimensional monolayers. The formation of a monolayer is the result of a compromise between the process of self-organization on the surface and the probabilities of spatial configurations of adsorbate molecules near the surface. The main reasons for the geometric heterogeneity of the monolayer are the geometric disorder and the energy inhomogeneity of the surface profile. A monolayer is a statistically related system and its symmetry causes correlations of processes at different spatial scales. The classical van der Waals equation is written for the two-dimensional, completely symmetric Euclidean space. In the general case, the geometry of the monolayer must be defined for the Euclidean space of fractional dimension (fractal space) with symmetry breaking. In this case, the application of the classical van der Waals equation is limited. Considering the fractal nature of the monolayer–solid interface, a quasi-two-dimensional van der Waals equation is developed. The application of the equation to experimental data of an activated carbon is shown.

**Keywords:** monolayer; van der Waals equation; fractal dimension; activated carbon; isotherm

## 1. Introduction

The concept of a monomolecular layer is widely used in thermodynamics [1,2]. Typically, one has in mind an adsorption layer of some substance at the interface of any material. The molecules forming the monolayer create some surface (two-dimensional) pressure [3]. The connection between the two-dimensional pressure and other parameters of the monolayer (density of molecules, temperature) is determined by the equation of state. The first two-dimensional equation of the state for the monolayer at low concentrations has been proposed by Traube [4]. This equation describes some ideal two-dimensional monolayer: $\pi A = RT$, where $\pi$ is the two-dimensional pressure and A is the area per one molecule [5]. De Boer [6] proposed an empirical equation of state $\pi(A-b_2) = iRT$, where parameter $b_2$ denotes the area per single molecule in a tightly packed layer, i is a measure of the attractive forces.

To date, different equations have been proposed for two-dimensional systems: power-type equations based on the virial expansions, the Frumkin logarithmic equation, and the Van der Waals (VdW) equation [7].

Recently, a new two-dimensional equation of the state of the monolayer was proposed [3]:

$$\overline{\pi} = \int_\theta \left(1 - \theta \frac{dh}{d\theta}\right) \frac{d\theta}{1 - \theta f^{ex}} + \Delta\pi \tag{1}$$

The first term under the integral takes into account the orientation effect (h is the dimensionless thickness of the monolayer) of asymmetric molecules. The value $\Delta\pi$ depends on the type of molecular

interaction, $\overline{\pi}$ corresponds to the dimensionless two-dimensional pressure, and $\theta$ to the degree of filling the surface. Molecular interactions are always calculated through the centers of mass of particles, and therefore two-dimensional pressure is determined not by the real area of the monolayer, but by the size of the area in which the centers of mass of the particles move. This means that the excluded area $a_{ex}$ must be understood as the area that is inaccessible to the centers of mass. Similarly, a particle with a center of mass at rest (but rotating along the surface and therefore round) creates an excluded area in the form of a circle with a radius equal to the sum of the radii of the particle at rest and the moving particle approaching it. If all particles are of the same size and move independently (the case of a one-component two-dimensional gas), then a stationary particle of radius $r$ creates an excluded area, which is four times the landing area $a_0 = \pi(2r)^2$ of the particle. Taking this into account, the exclusion factor $f^{ex}$ as the ratio of excluded and landing areas: $f^{ex} = \frac{a_{ex}}{a_0}$ is introduced.

For a monolayer of neutral, spherically symmetric molecules, Equation (1) has the form:

$$\overline{\pi} = \int_{\theta} \frac{d\theta}{1 - \theta f^{ex}} - \alpha' \theta^2 \tag{2}$$

The integration of Equation (2) depends on the choice of the exclusion factor $f^{ex}$ and $\alpha'$ is the dimensionless constant of intermolecular interaction.

If $f^{ex}$ = constant, the integration of Equation (2) leads to the Frumkin and Planck equation [8,9]:

$$\overline{\pi} = -\frac{\ln(1 - \theta f^{ex})}{f^{ex}} - \alpha' \theta^2 \tag{3}$$

When $f^{ex}$ = 4 this is a two-dimensional analogue of the Planck equation, and for $f^{ex}$ = 1 this is the Frumkin equation [3]. The definition of $\overline{\pi}$ will be given below.

As a first approximation, the exclusion factor $f^{ex}$ was chosen in the form [3]:

$$f^{ex} = 4(1 - \theta) \tag{4}$$

Then the integration of Equation (2) with (4) yields the two-dimensional equation of VdW

$$\overline{\pi} = \frac{\theta}{1 - 2\theta} - \alpha' \theta^2 \tag{5}$$

As a next approximation, the $f^{ex}$ dependence [3] was proposed:

$$f^{ex} = \frac{4(1 - \theta)}{1 + \kappa\theta} \tag{6}$$

where $\kappa$ is a constant ($\kappa \sim 1.144$).

Then, the integration of Equation (2) with (6) yields the two-dimensional equation [3]:

$$\overline{\pi} = \frac{2\beta + 4}{\beta^2} \ln(1 + \beta\theta) - \frac{\beta + 4}{\beta} \frac{\theta}{1 + \beta\theta} - \alpha'\theta^2 \tag{7}$$

The parameter $\beta$ is a quantity that depends on the excluded area: $\beta = -1428$. The correlations between $\kappa$ and $\beta$ are described in Rusanov [3].

Equation (7) allows us to describe the first three virial coefficients quite accurately.

Although such equations are usually considered two-dimensional, the three-dimensional aspect of real monolayers is inevitably present. For example, this is due to the influence of the surrounding bulk phase, the change of the orientation of non-spherical molecules in the monolayer with the change of the two-dimensional pressure. Therefore, generally speaking, the equation of state for the monolayer should be considered quasi-two-dimensional.

The description of the surface in terms of two dimensions is certainly appropriate if bumps and dips in the surface have extensions much larger than the size of any molecule. Such is no longer true when the irregularities of the surface are comparable to the size of adsorbate molecules. Consequently, to describe the irregular surface one should consider its dimension as an appropriate value between two and three.

Let us now explain the determination of the surface (two-dimensional) pressure. The monolayer is a real, in general, three-dimensional medium under the influence of molecular interactions. The influence of external fields will not be considered. A monolayer of spherically symmetric, neutral molecules on a geometrically and energetically homogeneous surface corresponds to a Euclidean space.

The mechanical state, the surface pressure of the monolayer, is determined by specifying a stress tensor or a pressure tensor. These tensors differ only in sign. For molecular systems, the pressure tensor is more generally accepted. In the general case (Adsorption on a fractal surface) the conformal conditions for a monolayer on a surface are not satisfied. Therefore, it is not possible to determine two-dimensional pressure in the framework of the classical Gibbs definition. Therefore, the authors use the definition of surface pressure as: $\pi = \left(\frac{\partial F}{\partial A}\right)_{N,T}$ with free energy $F$.

Two tangential components of the tensor of excess surface pressure, in general, determine the surface two-dimensional pressure. The task of strictly calculating the tensor components of the surface pressure is a very difficult one and practically impossible for the analysis of real (non-model) adsorption systems. Therefore, the only possible version for the calculation of surface pressure is the calculation of the value from the pressure of the experimental adsorption isotherm [6].

To correctly describe the state of a monolayer in a two-dimensional Euclidean space in a wide range of parameter changes, various approximations must be made. These approximations are not always strictly justified. When describing unordered media by the methods of fractal geometry, one fundamental assumption is made, namely the hypothesis of scale-invariant structures. This hypothesis allows us to construct a theory of disordered media using renormalization group methods. This approach is universal, but is associated with large mathematical (computational) difficulties. These difficulties were overcome by Mandelbrot's introduction of the concepts of fractal geometry. Thus, the description of the thermodynamic system, a quasi-two-dimensional layer of molecules, contains almost no assumptions or approximations.

The aim of this work is the modification of the two-dimensional VdW equation to a quasi-two-dimensional one. This procedure is based on the concept of the fractal nature of the monolayer–solid interface.

## 2. Theory

The surface pressure is generally defined as work required to change the surface area of the adsorbed phase on unit value at constant $V$, $N$, and $T$.

The definition of the surface pressure through the grand partition function is as follows [10]:

$$\frac{\pi A}{RT} = \ln \Xi^*$$
(8)

where $\Xi* = \Xi/\Xi^0$, $\Xi^0$ is the grand partition function for not adsorbed gas in the bulk phase.

The virial expansion method is theoretically exact specifically in the considered case of ethane adsorption on activated carbon. The logarithm of the grand partition function can be expanded in a series [10]:

$$\ln \Xi = Z_1^{(2D)} Z^{(2D)} + \left[Z_2^{(2D)} - Z_1^{(2D)}\right]^2 \times \left[\left(\frac{Z^{(2D)}}{2}\right)^2\right] + \left[Z_3^{(2D)} - 3Z_2^{(2D)} Z_1^{(2D)} + 2\left(Z_1^{(2D)}\right)^2\right]\left(Z^{2D}\right)^3 /6 + ..$$
(9)

Then, the equation for the surface pressure can be obtained by the substitution of the expansion in Equation (9) into Equation (8) [10]:

$$\frac{\pi A}{RT} = 1 + B_{2D}(N_a/A) + C_{2D}(N_a/A)^2 + ..\tag{10}$$

where $N_a$ is the number of the adsorbed gas molecules and $B_{2D}$, $C_{2D}$, ... are the two-dimensional virial coefficients. These coefficients are also determined for the three-dimensional case. For a large $N$, it is impossible to calculate $Z_N^{2D}$ exactly. Model approaches are necessary. However, the difficulties of calculation of the higher virial coefficients limit the applicability of this method by small values of the surface density of matter [10].

The equation for the surface pressure isotherm at high densities can be obtained as the two-dimensional VdW equation. In the thermodynamics of adsorption, it is more convenient to use the canonical partition function [10]. The equation that relates the canonical partition function with two-dimensional (surface) pressure has the form [10]:

$$\frac{\pi}{RT} = \left(\frac{\partial}{\partial A} \ln Z_N^S\right)_{T,V_a,N}\tag{11}$$

where the configuration interval is given by: $Z_N^S = Z_N^{(2D)}\left(Z_1^S\right)^{-N}$.

Let us give some clarifications concerning the value $V_a$. The whole volume $V$ of the gas phase in a thermostat can be divided into two parts: (1) a small region $V_a$ near the surface of the adsorbent, where the gas density differs substantially from the density in the bulk phase; (2) the part of the volume $V_d$, in which gas properties do not differ from the properties of the gas in the main volume.

The configuration integral $Z_N^S$ can be represented in two parts:

(1) three-dimensional integrals $Z_1^S$ for an individual molecule on the surface;
(2) a configuration integral $Z_N^{2D}$ in which the integration is carried out in the plane of the parallel surface.

The configuration integral $Z_1^S$ is calculated by Equation (12):

$$Z_1^S = A z_f^s \exp(-\varepsilon_1^s/RT)\tag{12}$$

In Equation (12) $z_f^s$, the amplitude of the free oscillations of the molecule is perpendicular to the surface; $\varepsilon_1^s$ is the magnitude of the energy of the molecule at the minimum of the potential well. The integral $Z_N^{2D}$ is defined as follows [10]:

$$Z_N^{2D} = A^{-N} \int_A \cdots \int_A - \sum_{1 \le i < j \le N} [u_s(r_{ij})/RT]dr_1 \ldots d_N\tag{13}$$

In Equation (13), $u_s(r_{ij})$ is the energy of interaction of the $i$-th molecule with the $j$-th molecule. In case of moving (or restrictedly moving) films (here: adsorption layers) the dependence $u_s(r_{ij})$ on $r_{ij}$ can be neglected. This approximation is valid for the isotherm under consideration. With the Lennard-Jones potential as the interaction potential of the molecules, the integral in Equation (13) is transformed to the form:

$$Z_N^{2D} = \left[(A_f/A)exp\left(-\frac{\phi_0}{RT}\right)\right]^N\tag{14}$$

In Equation (14), $\phi_0$ is the average energy of lateral interactions per molecule. Then combining Equations (12) and (14), the configuration integral $Z_N^S$ will be written in the form:

$$Z_N^S = \left[\left(A_f z_f^s\right)exp\left(-\frac{\phi}{2RT}\right)\right]^N \tag{15}$$

where $A_f$ is an excluded area of the molecule in the adsorption layer, $\phi$ is the average energy of intermolecular interactions in the potential field of the adsorbent. In Equation (15), it is assumed that

$$A_f = A - b_2 \tag{16}$$

$$\frac{\phi}{2} = -\frac{Na_2}{A} \tag{17}$$

$$b_2 = \pi\sigma^2/2 \tag{18}$$

$$a_2 = \pi\varepsilon\sigma^2 \tag{19}$$

where $\sigma$ (molecular diameter) and $\varepsilon$ (depth of potential well) are the parameters of the Lennard-Jones potential for molecules in the monolayer [6].

Let us give some comments concerning the parameter $b_2$. According to Equation (18), the value of the parameter $b_2$ depends only on the molecular diameter $\sigma$. However, in real adsorption monolayers, the parameter $b_2$ is defined as the area accessible for the molecule at the maximum density of the monolayer. The maximum density of the monolayer is determined by the type of molecules, by the properties of adsorbent and by the temperature. Let us denote the number of molecules in the monolayer at maximum filling for a given temperature as $N_m$; the number of molecules adsorbed at given temperature and pressure $p$ is denoted as $N$. In this case, the number of sites available for adsorption will be equal to $N_m$-$N$. The number of molecules $N$ and $N_m$ are associated with the areas occupied by a molecule by means of relations [6]:

$$N \propto 1/A \tag{20}$$

$$N_m \propto 1/b_2 \tag{21}$$

Usually, the values $N_m$ and $N$ are determined experimentally as the number of moles per unit weight of the adsorbate and are denoted as $V_m$ and $V$ respectively. Then, from Equations (20) and (21), we obtain the area occupied by one molecule at an arbitrary $p$ in a dense monolayer:

$$A = \frac{S_{exp}}{VN} \tag{22}$$

$$b_2 = \frac{S_{exp}}{V_m N} \tag{23}$$

where $V$ and $V_m$ are the adsorption values at pressure $p$ and in maximum dense monolayer, respectively.

In the Equations (22) and (23), $S_{exp}$ is the specific surface area of adsorbent. Thus, in future, we determine the value $b_2$ for each individual isotherm by Formula (23). The same conclusions must be made concerning parameter $a_2$, as well. The parameters $b_2$ and $a_2$ are determined from the experimental isotherm in each case separately.

Substituting Equations (15)–(17) into Equation (11), we obtain the well-known VdW equation:

$$\pi = \frac{RT}{A - b_2} - \frac{a_2}{A^2} \tag{24}$$

Then, substituting Equations (22) and (23) into Equation (24), we rewrite Equation (24) in the form:

$$\pi = \frac{RTV_mN}{S_{exp}} \frac{\theta}{1-\theta} - a_2\left(\frac{V_mN}{S_{exp}}\right)^2\theta^2$$
$$\text{with}: \ \theta = \frac{V}{V_m} \tag{25}$$

It is also convenient to consider the equation of state of a monolayer in the dimensionless form:

$$\overline{\pi}(T,\theta) = \frac{\pi}{RTV_m} \tag{26}$$

In this case, the strictly two-dimensional equation of state has the form:

$$\overline{\pi}(T,\theta) = \frac{\theta}{1-\theta} - a_2'\theta^2$$
$$\text{with}: \ a_2' = \frac{a_2V_mN}{RTS_{exp}} \tag{27}$$

It should be noted that the Equations (16) and (17) are obtained for a homogeneous surface. The topological dimension of the interface between the monolayer and the surface of the adsorbent is equal to two. For inhomogeneous adsorbent surfaces, the dimension of the interface between the monolayer and the surface is defined as fractal, with a fractal dimension $D_F$ with $D_F > 2$ [11].

Now, consider the VdW equation for a fractal surface. In order to correctly apply Equations (16) and (17) to the fractal surface, it is necessary to take into consideration the effect of the fractal dimension of the surface on the value of area occupied by a molecule. Consider two types of adsorbent surface: homogenous, with the topological dimension $D = 2$, and rough, with the fractal dimension $D_F > 2$. On each surface, let us define an area bounded by the perimeter L. Suppose that the perimeters for homogeneous and rough surfaces are the same. Then the following equality is fulfilled [12]:

$$A_F^{1/D_F} = A^{1/2}$$
$$A_F = A^{(D_F/2)} \tag{28}$$

In the future, we will use the notation $\alpha = D_F/2$. Accordingly, the value of the excluded area of the molecule in the monolayer on the fractal surface will be equal to:

$$A_{f,F} = A^\alpha - b_{2,F}^\alpha$$

Now it is necessary to rewrite Equations (16) and (17), taking into account Equations (28). Then the integral for the monolayer on the fractal surface should be written as:

$$Z_{N,D}^{2D} = \left[\frac{A_F^\alpha - (b_{2,D})^\alpha}{(b_{2,D})^\alpha}exp\left(-\frac{a_{2,D}}{2RTA^\alpha}\right)\right]^N \tag{29}$$

Substituting Equation (29) into (11) we obtain:

$$\pi_F = \alpha\frac{RTV_mN}{S_{exp}}\frac{\theta}{1-\theta^\alpha} - \alpha a_{2,D}\left(\frac{V_mN}{S_{exp}}\right)^\alpha\theta^{1+\alpha} \tag{30}$$

It is also convenient to consider the equation of state of a monolayer in the dimensionless form:

$$\overline{\pi}_F(T,\theta) = \alpha\left(\frac{\theta}{1-\theta^\alpha} - a_{2,D}'\theta^{1+\alpha}\right);$$
$$a_{2,D}' = \frac{a_{2,D}}{RT}\left(\frac{V_mN}{S_{exp}}\right)^{\alpha-1} \tag{31}$$

The strictly two-dimensional equation of state, Equation (27), contains one parameter $a_2'$, whereas the quasi-two-dimensional equation of state, Equation (31), contains two parameters: $\alpha$ and $a_{2,D}'$.

In deriving Equation (30), the influence of the degree of disorder of the adsorbent (fractal dimension) on the properties of Equation (30) is strictly shown. In this case, the dependence of the excluded volume (the first term of Equation (30)) and of the molecular interaction (the second term of Equation (30)) on the fractal dimension $D_F$ is strictly obtained. If all the states of the adsorption medium are equally probable, $D_F = 2$ and Equation (30) reduces to the classical equation of van der Waals.

## 3. Results and Discussion

As an example, consider the adsorption isotherms $a(T, p)$ of ethane at $T = 273.15$ K (sample №1) and $T = 323.15$ K (sample №2) on activated carbon sample with specific surface $S_{BET} = 1440$ m²/g (Figure 1). In Figure 1, the points denote the experimental values [13,14] and the lines are computer fittings to the Toth equation.

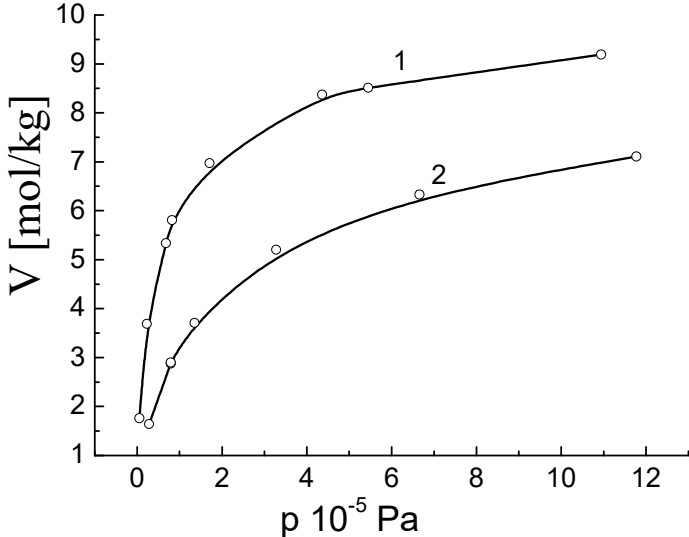

**Figure 1.** Adsorption isotherms of ethane on KF-1500 at 273.15 K (sample №1) and at 323.15 K (sample №2).

The isotherms of the surface pressure $\pi(T, p)$ are calculated using the adsorption isotherms as follows [6]:

$$\pi = RT \int_p \frac{a(T,p)}{p} \, dp \tag{32}$$

Further, we will determine the value of the surface pressure $\pi$, calculated by Formula (25), as the experimental value and denote as $\pi_{exp}$. We assume that we do not know *a priori* the equation of the adsorption isotherm $a(T, p)$, but we use Equations (25) and (30) with free parameters for alpha, $a_2$, and $a_{2,D}$. The surface pressure $\pi(T,p)$ will be calculated by Equation (32) using Newton-Kotes numerical integration formulas [15]. For further analysis, it is necessary to represent the surface pressure isotherm $\pi(T,p)$ in the form $\pi(T,\theta)$, where $\theta = V/V_m$, where the parameter $V_m$ is determined by the adsorption Toth isotherm $a(T, p)$ [6].

For the adsorption isotherm of ethane on activated carbon (S = 1440 m²/g) at $T = 273.15$ K (sample №1) the following values were obtained: $a'_2 = 0.87$; $a'_{2,D} = 2.1$; $\alpha = 1.2$. For sample 2: $a'_2 = 0.85$; $a'_{2,D} = 1.75$; $\alpha = 1.14$. The results of comparative graphical analysis of the calculated and experimental data are presented in the form of graphs:

$$Y = \frac{\overline{\overline{\pi}}_{2,cal}}{\overline{\overline{\pi}}_{2,exp}} = f(\theta) \tag{33}$$

Figure 2 represents the results of analysis for sample №1 and Figure 3 for sample №2. Figures 2 and 3 are constructed as follows. Ordinates of quantities $Y$ are calculated from Equation (33) of the knots of the function $V(p)$. As the knots of the function, the values of the argument (the value of the pressure) at which the adsorption was measured were chosen.

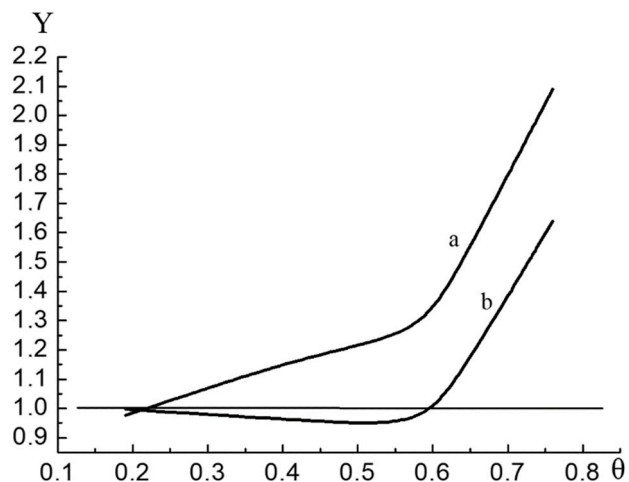

**Figure 2.** Results of analysis for sample №1 with Formula (33). (a) Equation (27); (b) Equation (31).

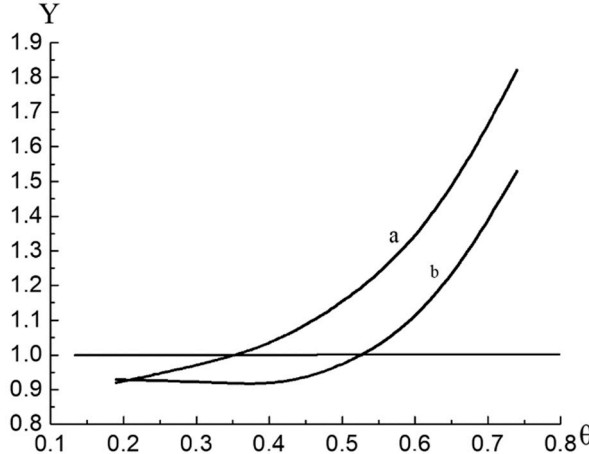

**Figure 3.** Results of analysis for sample 2 with Equation (33). (a) Equation (27); (b) Equation (31).

For sample 1, Equation (27) describes the experimental surface pressure isotherm for $0 < \theta \le 0.35$ with a maximum relative deviation of 9.5% and for sample for $0 < \theta \le 0.45$ with a maximum relative deviation of 9.7%.

The quasi-two-dimensional Equation (31) describes the experimental surface pressure isotherm for sample №1 for $0 < \theta \le 0.65$ with a maximum relative deviation of 8.5% and for sample 2 for $0 < \theta \le 0.67$ with a maximum relative deviation of 8.7%.

Earlier, the designation $\alpha = \frac{D_F}{2}$ was introduced. Consequently, the fractal dimension of the interface between the monolayer and the surface of the adsorbent is $D_F = 2.4$ for sample 1 and is $D_F = 2.28$ for sample 2. It is interesting to check the results of the calculation of these fractal dimensions obtained with the quasi-two-dimensional Equation (31). In earlier published work, the Langmuir adsorption isotherm equation has been investigated in the form [16]:

$$V = V_m \frac{(bp)^\beta}{1 + (bp)^\beta} \tag{34}$$

In Equation (34), $V$ and $V_m$ are the adsorption values at pressure $p$ and in the maximum dense monolayer, respectively; $b$ is an equilibrium constant for $p \to 1$; $0 < \beta \leq 1$. If $\beta = 1$, Equation (34) turns into the Langmuir equation corresponding to the adsorption on a homogeneous surface without taking into account the interaction between the adsorbate molecules. It has also been proven rigorously that the parameter $\beta$ and fractal dimension $D_F$ of the interface between the monolayer and the adsorbent surface are related by the equality: $D_F = 3 - \beta$ [11].

According to Equation (34), sample 1 is characterized by the parameter value $\beta = 0.68$. For sample 2, we found $\beta = 0.76$. Consequently, the fractal dimension of the interface between the monolayer and the adsorbent surface for sample 1 is $D_F = 2.32$ and for sample 2 is $D_F = 2.24$. These results agree well with the values of the fractal dimension determined from quasi-two-dimensional Equation (31).

Now let us give some comments regarding the interval of θ, for which Equation (31) describes the surface pressure isotherm. To do this, it is convenient to analyze the isotherms for the Graham functions [17]:

$$Gr(T, \theta) = \frac{\theta}{1 - \theta} \frac{1}{p} \tag{35}$$

Figure 4 shows the function $Gr(T, \theta)$ of the adsorption isotherm for sample №1. The dots denote the values calculated by Formula (35) based on the experimental adsorption isotherm.

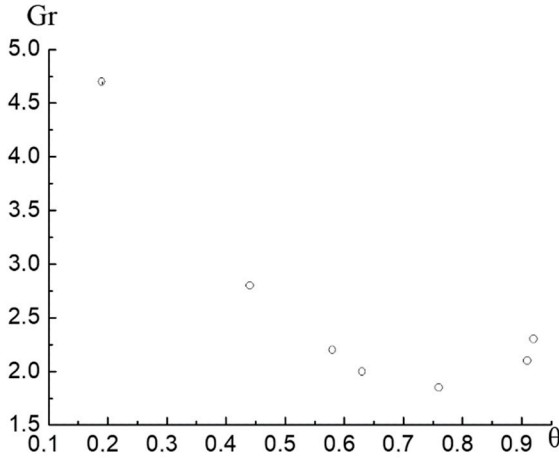

**Figure 4.** Function $Gr(T, \theta)$ of the adsorption isotherm of sample №1.

Such behavior of $Gr(T, \theta)$ is typical for the monolayer on non-uniform surface taking into account the interaction of adsorbed molecules. Figure 4 clearly shows that the function $Gr(T, \theta)$ is not monotonic: it reaches a minimum at $\theta \approx 0.65$ and then increases sharply. The domain of applicability of the quasi-two- dimensional VdW equation is limited by the monotonic domain of the function $Gr(T, \theta)$.

## 4. Conclusions

The further development of the two-dimensional van der Waals equation by considering the fractal nature of the monolayer–solid interface delivered a quasi-two-dimensional van der Waals equation. The symmetry of the monolayer determines the number of possible equally probable micro configurations of the macro state of the adsorbate. When adsorbed on a geometrically and energetically homogeneous surface, the probabilities of all configurations of the monolayer are the same and independent. The van der Waals equation is strictly valid for the region of the second and third virial coefficients. In this region, Equation (30) is an analytic function with respect to a variable. When adsorption on a geometrically and energetically inhomogeneous surface takes place, the symmetry of the monolayer is broken. In this case, the probability of configurations of the monolayer is not the same and there are conditional probabilities. However, in this case, individual configurations of the monolayer can be geometrically ordered based on the scale invariance hypothesis, which makes it

possible to increase substantially the domain of determination of the van der Waals equation. However, in this case, the van der Waals equation (Equation (30)) is a nonanalytic function with respect to a variable $\theta$.

Application of the developed equation to experimental data of two different samples of activated carbon demonstrated a good compliance of measured and calculated data. The determined fractal dimensions of the coal samples coincide with the values calculated by another method.

**Author Contributions:** Writing—review and editing, V.K. and E.S.; methodology, V.K.; formal analysis, V.K.; investigation, E.S. All authors have read and agreed to the published version of the manuscript.

**Funding:** This research received no external funding.

**Conflicts of Interest:** The authors declare no conflict of interest.

## Symbols

| | |
|---|---|
| $a$ | adsorbed amount |
| $a_2$ | parameter in Equation (19) |
| $A$ | molecular area |
| $A_f$ | excluded area of a molecule |
| $b$ | equilibrium constant (FHH-eq.) |
| $b_2$ | parameter of de Boer equation |
| $b_2$ | parameter in Equation (18) |
| $B_{2,D}$ | virial coefficient |
| $C_{2,D}$ | virial coefficient |
| $D_F$ | fractal dimension |
| $f^{ex}$ | exclusion factor |
| $F$ | free energy (Helmholtz) |
| $Gr$ | Graham function |
| $h$ | dimensionless thickness |
| $i$ | measure of attractive forces |
| $L$ | perimeter |
| $N$ | number of moles |
| $N_a$ | number of adsorbed molecules |
| $N_m$ | number of moles in monolayer |
| $p$ | pressure in bulk phase |
| $r_{ij}$ | distance between molecule i and j |
| $R$ | ideal gas constant |
| $S_{exp}$ | specific surface area |
| $T$ | temperature |
| $u_s(r_{ij})$ | interaction energy of molecule i and j |
| $V$ | adsorbed volume |
| $V_m$ | maximum of adsorbed volume |
| $V_a$ | gas volume where adsorbate exists |
| $V_d$ | gas volume with same density as bulk gas |
| $z_f^s$ | amplitude of oscillations |
| $Z_N^S$, $Z_N^{2D}$ | configuration integrals |
| $\alpha'$ | dimensionless constant |
| $\beta$ | parameter (Equation (7)) |
| $\varepsilon$ | depth of Lennard-Jones potential |
| $\pi$ | 2-dimensional pressure |
| $\overline{\pi}$ | dimensionless pressure |

| $\kappa$ | constant (Equation (6)) |
|---|---|
| $\sigma$ | molecular diameter (LJ-potential) |
| $\theta$ | degree of pore filling |
| $\phi_0$ | energy of lateral interactions (Equation (14)) |
| $\phi$ | average interaction energy (Equation (15)) |
| $\Xi$, $\Xi^0$, $\Xi^*$ | grand partition functions (Equation (8)) |

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
