# Peer review of "Van der Waals Equation for the Description of Monolayer Formation on Arbitrary Surfaces"

_colloids, doi:10.3390/colloids4010001_

Round 1
Reviewer 1 Report
The topic of the study is interesting enough. It should be mentioned also that this particular area of adsorption science, and also the approach used are comparatively seldom in recent literature. On my side, I would especially welcome this publication.
However, some corrections are necessary.
The title of Section 2 “Materials and Methods” is inappropriate. The submission is mainly theoretical; I propose “Theory” instead. It is necessary to re-plot also the experimental data in the coordinates used (Y vs θ) to be shown in Figs 2 and 3. The reader would expect the comparison between the theoretical curves and experimental values.3. Line 316: the correct spelling is “Fainerman” , not “Fainermann”
Author Response
Dear Reviewer,
thank you very much for your remarks.
The title of section 2 has been changed.
In Figure 2 and 3 only theoretical results are presented. The surface pressure isotherms were not studied experimentally.
The spelling in line 316 is corrected.
Best regards and Merry Christmas
The Authors
Reviewer 2 Report
The authors have developed a quasi-two dimensional Van der wall equation considering fractal nature of monolayer adsorption on solid surfaces. This equation’s validity was verified for adsorption on activated carbon surfaces.
The result itself is quite interesting, and worth publishing. It will be intriguing to investigate if this equation can shed more light on monolayer adsorption of surfaces other than carbon.
The paper is very well written, and reviewer has no major recommendation for change.
The authors may include following minor changes/further clarifications /insight:
Line 69: authors may want to include how orientations of non-spherical molecules may change two-dimensional pressure. Line 75: authors may want to clarify why one should consider dimension of irregular surfaces between two and three. Equation (13) formatting error in Us (rij ) line 159 : ( formatting error ) remove the period between V and respectively line 215: for consistency authors may want to write Figure 3 instead og Fig . 3Author Response
Dear Reviewer,
thank you very much for your remarks.
Line 69: The authors only want to include that orientations of non-spherical molecules may change two-dimensional pressure.
Line 75: The authors only want to clarify that only dimensions between two and three are interesting for irregular surfaces, because these surfaces are not totally flat like 2-dimensional objects and they are not volume-filling like 3-dimensional objects.
The mistakes in equation 13, in line 159 and 215 are improved.
Best regards and Merry Christmas
The Authors